# Alcohol Consumption in the Workplace: A Comparison between European Union Countries’ Policies

**DOI:** 10.3390/ijerph192416964

**Published:** 2022-12-17

**Authors:** Ivan Borrelli, Paolo Emilio Santoro, Maria Rosaria Gualano, Antongiulio Perrotta, Alessandra Daniele, Carlotta Amantea, Umberto Moscato

**Affiliations:** 1Department of Health Science and Public Health, Section of Occupational Health, Università Cattolica del Sacro Cuore, Largo Francesco Vito 1, 00168 Rome, Italy; 2Department of Woman and Child Health and Public Health, Fondazione Policlinico Universitario A. Gemelli IRCCS, 00168 Roma, Italy; 3School of Medicine, Saint Camillus International University of Health Sciences, 00131 Rome, Italy; 4Department of Prevention, U.O.S.T. Interdistrettuale Ambienti di Lavoro Ambito Sud, 84124 Salerno, Italy

**Keywords:** alcohol, European countries, health policy, occupational medicine, addiction

## Abstract

Background: Alcohol use is an ever-growing phenomenon in the population, consumption data indicate that 5–20% of the European working population have serious problems related to alcohol. The use of alcohol constitutes a risk to the health and safety of workers as well as to the safety of the general population. The present work aims to address the problem of alcohol intake in occupational settings by comparatively analyzing alcohol consumption behavior within the 27 countries of the European Union. Methods: The purpose of this research is to analyze the differences between the 27 countries of the European Union in the application of measures to assess and manage the risk of alcohol intake in occupational settings. Results: An examination of the legislation and guidelines of the different countries reveals profound differences in the management of the problem of alcohol in the workplace. The discrepancy is very wide that it ranges from the complete absence of legislative restrictions on a national level in some countries to highly restrictive measures with severe sanctions in others. Conclusions: It would be appropriate—also for the purpose of ease of movement of workers within the European Community—to find shared management models useful for protecting the health and safety of workers and the general population.

## 1. Introduction

The use of alcohol while working is a significant phenomenon that constitutes an additional risk to pre-existing occupational hazards; it is estimated that between 5% and 20% of the working population in Europe have serious problems related to their alcohol usage [1].

Alcohol use is an important issue in the workplace since it can lead to an increase in accidents, injuries, absenteeism, and inappropriate behavior by reducing a worker’s psycho-physical integrity, and significantly affecting the health and safety of third parties [2,3,4,5].

Workers under the influence of alcohol can be a danger to others, and therefore a risk factor, if they are working under the influence of alcohol, especially in occupational activities which involve a high risk of injury at work (e.g., police force, construction, transport, etc.) [6]. Although total alcohol consumption per capita in Europe fell from 12.3 L in 2005 to 9.8 L in 2016 [7], it remains among the highest in the world [8].

According to Eurostat statistics, in 2019, 8.4% of the adult working-age population in the EU consumed alcohol daily; 28.8% weekly; and 22.8% monthly. Daily alcohol intake was most frequent in Portugal, with 20.7%, followed by Spain with 13.0%, and Italy with 12.1%. In the Netherlands, almost half the population, 47.3%, consumed alcohol weekly, followed by Luxembourg with 43.1% and Belgium with 40.8%. By comparison, monthly consumption in the EU was highest in Lithuania at 31.3%. In all EU Member States, men consume alcohol more frequently than women [9]. 

Alcohol-related harms are a severe public health problem, accounting for over 7% of illnesses or premature death in the EU. Even moderate alcohol consumption increases the long-term risk of heart disease, liver disease, and cancer, and frequent consumption of large amounts can lead to addiction [4].

Approximately 800 people die every day in Europe from causes attributable to alcohol consumption, largely and predominantly from cancer (29%), liver cirrhosis (20%), cardiovascular diseases (19%), and injuries (18%). Alcohol-related harm not only affects users; increasingly, the consequences of alcohol consumption also directly impact families and the community at large through the deterioration of personal and work relationships, criminal behavior, loss of productivity, and health care costs [10].

Alcohol abuse can have a negative impact on work environments and employee performance [6].

Many studies have found a significant correlation between workplace stress and high levels of alcohol consumption [11,12,13]. Alcohol use has been shown to increase the risk of unemployment and, for those employed, absenteeism [5]. It has also been found to increase the risk of tardiness at work and/or leaving work early, resulting in loss of productivity; the development of inappropriate behavior, such as theft and other crimes; poor relations with colleagues; and low company morale [1,3,5,13]. In general, productivity loss costs are the dominant feature of studies on the social costs of alcohol-related harm, accounting for approximately one half of the total social cost of alcohol use in the EU [14]. 

Alcohol consumption is higher in some production sectors. These include, according to the authors Benavides F. et al., the food service sector, the alcohol trade, and the construction industry [15]. 

However, no type of work is exempt from the phenomenon, for example, the review by Virtanen M. et al. shows that regardless of the sector, individuals whose working hours exceed standard recommendations are more likely to exhibit increased levels of alcohol consumption that pose a health risk [16].

An explanation behind the causes for the increase in this phenomenon requires careful psychological, social, and even legal considerations.

Employers have a duty under health and safety laws to protect the health, safety, and welfare of employees and others affected by their activities.

Furthermore, workplaces can provide opportunities for health education and have a potential role in supporting the social reintegration of people with addiction issues.

This review aimed to perform research on workplace legislation concerning alcohol consumption, testing, and occupational consequences. 

The study was performed on all 27 EU member countries, to appraise the differences between different geographical areas, as well as cultures and socio-economic environments.

This work is part of a series of European and global regulatory reviews conducted by our research group [17,18,19].

## 2. Materials and Methods

A legislative review was carried out for all 27 EU Member States. A narrative review methodology was adopted to screen workplace alcohol legislation through a “grey literature” search using the databases of European and Member State institutional repositories and the Google search engine. The search was subsequently extended to the scientific literature, using the PubMed database, when no specific regulations emerged from governmental website databases. To identify the national legislation in force in each country, the national archives were screened in their original language (labor code, civil code laws related to workplace safety, occupational health and safety laws, etc.), to highlight the policies on alcohol use in the workplace related to consumption, testing or consequences.

The scientific literature research conducted through PubMed—when required—involved selecting studies published over a 10-year period (January 2012–June 2022); for each EU country, studies mentioning alcohol and drug use in the workplace were included and screened for the relevant legislation mentioned in the paper.

## 3. Results

We analyzed the national legislation on alcohol in the workplace in the 27 countries of the European Union.

### 3.1. National Legislation in the 27 Countries of the European Union

#### 3.1.1. Austria

According to the Employee Protection Act [20] employees “may not use alcohol, medicines or narcotics to put themselves in a state in which they could endanger themselves or other people” (Section 15, Paragraph 4). Fines from EUR 218 to 360 may be enforced (Section 130, Paragraphs 4 and 5) if this prohibition is disregarded. There is no general ban on alcohol during working hours for employees subject to the Employee Protection Act; rather, it is only a relative ban that does not distinguish between alcohol consumption during or outside of working hours and particularly impacts the behavior of employees before starting work and during rest breaks. Alcohol bans or restrictions at the workplace can result from collective agreements and company agreements; they can also be agreed upon in the employment contract or ordered through instructions from the employee. For the construction industry, the Construction Worker Protection Ordinance [21] stipulates that workers impaired by alcohol, medicines, or narcotics are not allowed to enter the construction site. During working hours, the consumption of alcoholic beverages is prohibited, but during rest breaks, workers may consume these types of beverages if it is ensured that the workers do not put themselves in a condition in which they endanger themselves or other workers on the construction site. The Ordinance on the Implementation of the Motor Vehicles Act [22] stipulates that drivers of vehicles are not allowed to start work in a state impaired by any substances or to consume alcohol during the entire period of work. Since the term “operating time” is based on the Working Time Act-AZG [23], the consumption of alcohol is also prohibited during breaks and rest periods. However, employees are not obliged to undergo an alcohol or drug test without their expressed consent [24]. If an employer, based on reasonable doubt, requires an employee to undergo a test, the latter has the right to refuse this control measure, unless there are special provisions due to the elevated level of danger in the employee activity that are subject to special control (e.g., flight personnel). Nonetheless, the testing doctor, is unauthorized to transmit the result to employers due to medical confidentiality unless the employee expressly allows it [25]. Furthermore, dismissal for refusing the test is unjustified except in cases of a particularly serious nature (Section 82 lit b GewO 1859). If a worker is unable to work due to intoxication, the employer and/or supervisor must send them home, and the employee is not entitled to continued payment of wages for that shift, since it is considered an unexcused absence. The circumstances differ for employees suffering from alcohol addiction who retain their right to continued payment of wages. Pursuant to Section 82 of the Trade Ordinance, GewO [26], a worker may be dismissed immediately if he has been repeatedly warned to no avail. The legislation requires that the employer must intervene at least two times, admonishing the workers, to fulfill the grounds for dismissal due to intoxication.

#### 3.1.2. Belgium

The most important Labor laws relevant to alcohol testing in Belgium are the General Regulation for the Protection of Labor (RGPT) [27] and the Act respecting contracts of employment [28]. According to the General Regulation for the Protection of Labor and Act 28 of January 2003 [29] medical testing, for example, is authorized pursuant to jobs involving certain risks, for instance, in safety-sensitive positions or in positions which involve exposure to an occupational health hazard. Individuals applying for these types of jobs are also required to undergo pre-employment screening. The conduction of alcohol testing constitutes an invasion of privacy; a worker who is not performing job tasks considered to involve risk must express consent to be tested. The welfare law of 4 August 1996 [30] related to the well-being of workers during their work, obliges the employer to take necessary measures to promote the well-being of workers while performing their work. In the Belgian private sector, collective agreement number 100 [31] was signed on 1 April 2009 and set out specific regulations on alcohol and drug use. These tests must comply with several conditions to retain their validity. No biological or medical tests may be used (breath testing is considered non-medical). Only tests that give no exact percentage of intoxication, but only a positive or negative indication of intoxication (such as breathalyzers or psychomotor tests) are permitted. Moreover, tests cannot be used in isolation but must be part of a set of policy implementation measures [28]. The right to privacy must be preserved, as set out in Article 8 of the European Convention on Human Rights, Article 17 of the International Treaty on Civil and Political Rights, and 22 of the Constitution. However, the right to privacy is not absolute. Drinking alcohol in the workplace does not necessarily lead to dismissal for cause; the court and tribunal examine the circumstances under which the alcoholic consumption took place.

#### 3.1.3. Bulgaria

In Bulgaria, alcohol consumption at work is subject to the Labor Code [32]. According to Article 187 of the Labor Code, it is an offense against labor discipline for an employee to “report to work in a condition that prevents him from performing the tasks assigned to him”. Therefore, the duties of employees described in Article 126 include the employee’s obligation to “report to work in a condition that enables him or her to perform the assigned tasks and shall not use alcohol or any other intoxicating substance during working hours”. This means that the employer has the right to impose disciplinary action, which may include a reprimand, a dismissal warning, or a dismissal. Article 199 states that “the employer or the immediate supervisor may suspend from work an employee who appears in a condition that prevents him from performing his work duties, uses alcohol or any other strong intoxicant during working hours”. The suspension continues until the employee is again considered fit for duty. Additionally, the employee does not receive payment for that period. The Labor Code makes no reference to alcohol and drug testing and there are no established rules on exactly how the condition of the employee is established and similarly, how it is assessed that the worker has regained his or her fitness. Specific decrees on mandatory alcohol screening in the workplace apply only to certain categories of transport workers before the start of their working day (pre-shift instructions) [33].

#### 3.1.4. Croatia

In Croatia, in accordance with Section 58, paragraph 5 of the Occupational Safety and Health Act (Official Gazette, Nos. 71/14, 118/14, 94/18, and 96/18) [34], the employer is required to establish in writing the procedure which is to be followed to verify whether the worker is under the influence of alcohol or other intoxicating substances. The verification procedure is conducted with the consent of the worker. Furthermore, the method of verification, the type of test or device, the recording and the confirmation method of the results, as well as the procedure in case of refusal of the worker to access the verification, must be determined. The worker is tested for alcohol or other addictive substances through a breathalyzer or other appropriate device, procedure, or means [35]. The verification procedure mentioned above is regulated by the employer in the ordinance or another act (in written form), and since the method of training is not prescribed by the Ministry of Labor and Pension System, it is the employer himself who determines how the persons who will carry out the verification will be trained, as well as the method of recording and confirming the results, and the procedure in the event of the employee’s refusal to access the verification. Article 64, paragraph 1 of the Act stipulates that to determine or verify the health capacity to perform certain tasks, based on the risk assessment, at his own expense the employer may subject the employee to a medical examination by an occupational health specialist before and during the employment relationship [36]. The regulations on jobs with special working conditions identify alcoholism and other addictions as contraindications for conducting certain dangerous work tasks (e.g., handling explosives, working at high altitudes, diving work) [37].

#### 3.1.5. Cyprus

In Cyprus, employers have a general duty to ensure health and safety at work under the Safety and Health at Work Law, 89(I)/199638, but there is no specific reference in regard to alcohol and drugs in the workplace, with the exception of a general prohibition of the use of controlled substances at the workplace. Section 15 “Duty of Employees” states that “every employee, while at work shall take reasonable care to ensure the safety and health of himself and of other persons who may be affected by his acts or omissions at work” and “no person shall intentionally or recklessly interfere with or misuse anything provided for the safety, health, and welfare of himself or other persons at work”. However, there is no provision in place regarding the verification of this regulation. Under this law, a company’s doctor may conduct certain tests with the sole purpose of ensuring that the health of an employee is not affected by the use or contact of dangerous substances. In some areas of employment, certain medical tests must be performed to ensure that a person is fit for work [38]. The employer must, in cooperation with its employees and their representatives, draw up a written company policy on the use of alcohol and other addictive substances in the workplace. In preparing this policy, the employer should, where possible, cooperate with specialists and experts in the field of substance dependence [39]. An employer can dismiss an employee without notice in the case of improper behavior in the course of his duties or serious or repeated violation and/or disregard of work regulations or other rules pertaining to the employment. If the employer does not exercise his right of dismissal within a reasonable period from the offense, the dismissal may be considered unlawful [40].

#### 3.1.6. Czech Republic

In the Czech Republic, there is a prohibition on the consumption of alcohol and other drinking substances at workplaces and during working hours, even outside of the workplace; it is mandatory not to enter the workplace under the influence of alcohol. The prohibition on consuming alcoholic beverages does not apply to employees working in adverse microclimatic conditions (e.g., foundries, glassworks, etc.), if consuming beer with reduced alcohol content (alcoholic content not exceeding 1.2% in volume), and for employees whose consumption of these beverages is part of the performance of work duties or is usually associated with the performance of these duties. Workers must be tested for alcohol or other addictive substances by a worker appointed by the employer [41]. In addition, Law No. 65/2017, Title III authorizes the employer to subject the employee, in case of suspicion of alcoholic beverages or drugs, to testing by authorized employees. In case of refusal, the employee must be sent for a professional medical examination. In the event of further refusal, the employee will be denied access to the workplace and will be subject to administrative sanctions [42].

#### 3.1.7. Denmark

In Denmark, there is no national legislation on alcohol and drug use in the workplace. Collective agreements prevail in private and public companies, and they are authorized to create their own alcohol and drug policies. If the company wants to systematically test employees for the influence of alcohol or other drugs, the control measures must comply with the 2006 LO and DA “Agreement on Control Measures” [43]. However, a collective agreement may also establish its own rules on control measures. If employers want to introduce control measures, there must be an operational justification and a sensible purpose. For example, the operation of dangerous machinery or a position which involves responsibility for the lives and safety of others (e.g., transportation). An employer cannot simply demand that an employee be tested, but there must be a reasonable doubt that justified the request. The test must also be conducted professionally and with minimal offence to the employee. Otherwise, it may constitute an abuse of the employer’s management rights. The employee may refuse the test. However, this refusal may become subject to scrutiny if a trade union case is subsequently initiated due to, a dismissal, for example. There are currently no best practice guidance notes, but the Danish Health Authority has issued information and guidance for alcohol policies and alcohol issues in the workplace [44].

#### 3.1.8. Estonia

In Estonia, according to Section 13. 15 of the Occupational Health and Safety Act, the employer must remove from the workplace an employee under the influence of alcohol, narcotic or toxic intoxication, or psychotropic substances. According to Section 14 of the same law, an employee is prohibited from working under the influence of alcohol, narcotic or toxic intoxication, or psychotropic substances. In practice, however, it is not always entirely clear how an intoxicated employee should be treated according to the requirements of the law. In Estonia, communication and access to test results are regulated according to the Personal Data Protection Act [45]. The employer must consider that the detection of an employee’s intoxicated state is, from the point of view of personal data protection, the processing of data related to the employee’s health, and this may only be conducted with the employee’s consent or in a case stipulated by law. The employer may terminate the employment contract in an emergency for a valid reason arising from the employee, due to which the continuation of the employment relationship cannot be provided for in accordance with the interests of both. Therefore, the law gives the employer the right and opportunity to terminate the employment contract in an emergency if the employee was under the influence at work despite the employer’s warning [46].

#### 3.1.9. Finland

In Finland, there are guidelines for the prevention of the use of alcohol and drugs in the workplace (prepared in tripartite negotiations with representatives of the government, central trade unions, and workers), such as the Act on the Protection of Privacy in Working Life 759/2004 [47], which includes regulations on drug and alcohol testing in the workplace based on the right of employers to process employees’ personal data. According to the legislation, healthcare workers, personnel with relevant laboratory training and health services must be utilized to conduct inspections and tests on the health of employees, and to take blood samples, as required by health legislation. Moreover, this legislation applies to alcohol and drug tests, but does not prevent alcohol tests from being conducted as breath tests (i.e., breath analyzer). In occupational health care, drug tests can be performed on an employee based on the Occupational Health Care Act 1383/2001 [48], according to which the necessity of a test is assessed by a health professional and not by the employer. The employer cannot request an alcohol test without the employee’s consent, and consent to testing may be previously outlined in the employment contract. The test must be performed by a health professional, except in the case of voluntary breath tests which may be performed by non-health professionals. In regard to the test result to be provided to the employer, only a general conclusion on an employee’s health can be made (“fit”, “fit with restrictions”, or “unfit”). In fact, the employer is not entitled to information on the results of these tests; he is only informed of an assessment of the employee’s ability to work for his task.

#### 3.1.10. France

According to the French Labor Code (Code du Travail), Articles R4228-20 [49] and R4228-21 [50], no alcoholic beverages, except for wine, beer, cider, or sherry, are allowed in the workplace; may be served in the company restaurant or at special events (e.g., farewell party, end-of-year party, etc.). No other alcohol is permitted. In the event of non-compliance with these regulations, a fine of EUR 10,000 may be imposed on each employee concerned. Moreover, it is forbidden to permit intoxicated persons to enter or remain in the workplace. In general, it is the employer’s responsibility to organize provision of services in order to avoid any negative impact on workers’ health (Code du Travail). This responsibility explicitly includes the prevention of alcohol and drug use at work (Article L4622-2) [51]. For example, if alcohol is present at a company party, the employer must ensure that the health of its employees is uncompromised, as they can be held liable if an intoxicated employee causes an accident due to their consumption of alcohol. The Labor Code does not allow employees to be systematically tested for alcohol/drug use, except for jobs that involve dangerous tasks for the employee or a third person; however, there is no detailed list of the jobs that fit into this criteria [52]. Only an occupational physician has the right to determine whether it is necessary to conduct alcohol tests on employees, without the obligation to communicate the results to the employer. However, the doctor will provide the employer with information on the suitability of the person concerned for work [52].

#### 3.1.11. Germany

In Germany, the use of alcohol and drugs in the workplace is not prohibited by law. Therefore, there is no legal blood alcohol limit for all employees, as for example, in the highway code. Under the Works Constitution Act (Betriebsverfassungsgesetz, BetrVG) [53], it is stipulated that management and work councils regulate their use. The only exceptions to the blood alcohol limit are very few occupational groups classified as relevant to public life safety or public road traffic. These are mainly bus and taxi drivers, other professional drivers, and pilots, for which testing is provided, regulated by the 2008 Regulation on Occupational Health Prevention (in German) [53]. Therefore, in most companies and public administration, it is solely at the discretion of employers as to what extent they tolerate alcohol consumption by their employees. The employer has the right to impose sanctions on the employee who refuses to take an alcohol test, which may result in loss of employment or suspension [53]. In Germany, regional and cultural particularities can be decisive; for example, in Lower and Upper Bavaria and in Franconia it is still common for many companies’ employees to have a glass of beer during the lunch break. If the consumption of alcohol is not expressly prohibited by the employment contract, it is up to the employee to decide whether and how much he or she wants to drink. However, if an accident at work occurs due to an employee’s consumption of alcohol, the employee may be sued and face legal repercussions.

#### 3.1.12. Hungary

Workplace drug tests are regulated by the “Resolution about Workplace Drug Tests” of 2005: The use of drug testing in the workplace should be based on constitutional principles and authorized by law, and not by the arbitrary decision or “coercion” of the employer, for the protection of workers’ rights and privacy [44]. Due to this decision, the Hungarian government has established which specific occupations and sectors may undergo alcohol or drug testing. These professions include healthcare workers, teachers, and the army or police workforce. When alcohol or drug testing is permissible by law, workers cannot refuse to undergo regular testing. Alcohol testing is typically initially performed using a breathalyzer test to evaluate acute alcohol consumption while on duty; if the breathalyzer test is positive, a blood test is subsequently required, to confirm the result, in a medical center or an accredited laboratory [45].

#### 3.1.13. Ireland

There is no legislation in Ireland that requires employers to test their employees for alcohol or drugs in the workplace. It remains the employer’s decision whether to test its employees and in most Irish companies it is not common practice. In fact, according to the Health, Safety and Welfare at Work Act 2005 [54], there are no legally binding regulations that oblige employers to test for drugs and alcohol in the workplace. However, the Health & Safety Authority (HSA) has published voluntary guidelines on drug testing at work. In the case of an employee suspected of intoxication by alcohol or drugs, the employer must assess whether the employee poses a risk of danger to self or to others. It is not required by law to conduct tests, but in the case of danger, the advised action is to remove the employee from the workplace.

#### 3.1.14. Italy

The Italian reference legislation on alcohol and alcohol-related problems is contained in Law 125/2001, which provides a set of measures aimed at prevention, treatment, and social reintegration of persons with alcohol addiction. Article 15 of the law [55] establishes a ban on consumption and administration of alcoholic and super-alcoholic beverages for workers assigned to tasks which might endanger the safety, security, and health of third parties, which are listed in the Provision of the Government-Regions Conference of 16 March 2006 [56]. Alcohol tests in the workplace may only be performed by occupational health physicians, through breathalyzer or blood test. Blood tests offer increased reliability but cannot be performed without the worker’s signed consent. A worker who abuses alcohol or who refuses to undergo alcohol tests (without due cause) may be subject to punishment by the employer, both criminally and disciplinarily. For some categories of workers, and not solely commercial drivers, the blood alcohol concentration limit is 0.00%, and the consumption of alcohol is forbidden in the workplace. Italian legislation aims to impose an absolute ban on drinking alcoholic beverages in the workplace, rather than to identify alcohol addiction in workers.

According to Art. 41 of Legislative Decree 81/08 [57], the health surveillance medical examinations must aim to verify the absence of alcohol addiction. Workers suffering from alcohol-related problems, who intend to access therapeutic and rehabilitation programs, may have the right to have their job security preserved for up to 3 years. The employer may not dismiss the worker, unless this is stipulated in the contract or, in any event, the conditions defined by law are not met. The recidivism of the conduct, especially when associated with drunkenness, is an element that could justify dismissal. 

#### 3.1.15. Latvia 

The legislation on alcohol and drugs in the workplace in Latvia is contained in the Council of Ministers Regulation No. 394 of 2008, which refers to the procedures for testing alcohol, narcotics, psychotropic, or toxic substances. This regulation states that employees under the influence of alcohol or drugs can be dismissed permanently. However, tests can only be carried out on workers if agreed upon in a collective agreement, an individual employment contract, and/or an internal working procedure. Moreover, tests must be performed by a specialized health professional and include an analysis of the alcohol concentration in exhaled air or a blood test. A copy of the results is given to the individual who requested the test to be initiated; another copy is kept at the testing institute for 5 years [9].

#### 3.1.16. Lithuania

Lithuania has amended its alcohol control legislation over the last 30 years, to reduce alcohol consumption in the country [58]. Working under the influence of alcohol and/or consumption of alcohol in the workplace is against the law in Lithuania. If an employee is found to be in violation of this law, the employer must ensure that the employee leaves the workplace for the remainder of the workday, suspending the worker without pay for the remaining duration of the shift. The Lithuanian Labor Code allows for the suspension of employees due to this misconduct; employers may suspend workers for 30 days with average pay, or for up to 3 months by requesting the suspension in writing to the competent authorities, addressing the reason for the suspension. In Lithuania, working under the influence of alcohol or drugs is considered a gross violation of contractual duties by the employee, who may therefore have their employment terminated [59]. The blood alcohol content (BAC) limit at work is 0.02%. An exception of 0.04% is made for employees whose duties are related to alcohol consumption (i.e., sommeliers) during their working hours. The BAC must be zero for workers driving during their duty (i.e., commercial drivers), and the employer is responsible for performing controls on these workers no more than 30 min before the start of the shift; if this is not possible, a medical visit must be conducted to assess whether the driver has consumed alcohol by a doctor appointed by the employer or in a medical facility. Controls may also be performed upon suspicion of misconduct. A “zero-tolerance” alcohol policy may also be implemented at a company level by the employer [60].

#### 3.1.17. Luxemburg

The consumption of alcohol and drugs in the workplace is not mentioned in Luxembourg’s Labor Code; however, in the Accident Insurance Association, Industrial Section, Clause 36 (Prescription 36 de l’Association d’Assurances Contre les Accidents, section industrielle), it is stated that the introduction and consumption of alcohol (with more than 10% volume) in the workplace are forbidden [61]. The employer may explicitly ban alcohol and drug use in the internal regulations of the company and monitor employees to enforce this ban. According to a Gran-Ducal Decree, medical examinations and tests on blood or urine for drug consumption may only be performed by the occupational physician [62]. No specific legislation in occupational categories exists to establish which workers must be subjected to testing, the decision is left to the employer through internal regulations, in compliance with the aforementioned Grand-Ducal Decree. No legislation exists concerning the type of substance tests that employers can use to monitor the internal ban [39].

#### 3.1.18. Poland

Several policies were introduced in Poland to reduce alcohol consumption and mortality in the country [63]. The Polish Labor Code states that drinking alcohol at work constitutes a serious breach of duties for the employee, who may therefore be fired without notice [64]. It is prohibited to consume alcohol at work and according to the Act on Upbringing in Sobriety and Counteracting Alcoholism, Art. 16, it is also prohibited to bring alcoholic beverages inside the workplace. Furthermore, Art. 17 stated that an employer may stop an employee from continuing to work if there is reasonable suspicion that the employee has consumed alcohol before or during work and require the employee to undergo a sobriety test; if the employee refuses, they may be considered intoxicated, in absence of a test [65]. In June 2019, the President of the Personal Data Protection Office issued an interpretation of the Entrepreneurs Law Act of 6 March 2018, stating: “The employee’s sobriety test may be carried out by the police, but not by the employer”, as the employee’s sobriety is considered health data, and may therefore only be initiated by the employee. This interpretation essentially means that the ability of the employer to perform sobriety checks on workers is blocked [66].

#### 3.1.19. Portugal 

Consumption of alcohol or drugs is not regulated by the Portuguese Labor Code [67] or in the Legal Regime for the Promotion of Health and Safety at Work [68]. Furthermore, drug or alcohol testing may only be performed by the occupational workers if the company establishes thorough internal policies specifying that controls must be conducted; however, the occupational physician may only inform the employer that the employee is not fit for duty without stating the explicit reason, as medical examination’s results cannot be disclosed by the occupational physician. If the company establishes internal policies that alcohol tests must be performed, a positive test result may result in a suspension or job loss. For the construction industry, the collective work contract states that random alcohol tests may be performed on all workers (or on workers that appear to be intoxicated during working hours) [44].

#### 3.1.20. Slovak Republic

The employer is responsible for testing employees for alcohol, narcotic drugs, or psychotropic substances during working hours and for compliance with the ban on smoking in the workplace. He/she shall identify the group of employees to be tested for the assessment of the use of alcoholic beverages or prohibited substances and the assessment of possible addiction. Employees are obliged to refrain from consuming alcoholic beverages, narcotic drugs, and psychotropic substances at the workplace and outside of the workplace during working hours. The employee is obliged to refrain from reporting to work under this influence and to undergo examinations organized by the employer or the competent state authority to ascertain whether he/she is under the influence of alcohol, narcotic, and/or psychotropic substances. The prohibition on consuming alcoholic beverages in and outside of the employer’s places and premises during working hours does not apply to employees for whom the exceptional consumption of alcoholic beverages is part of their work duties or is normally connected with the performance of these duties [68].

An employer is obliged to set a medical examination as a precondition of employment only if the applicant is a juvenile or for jobs where a special legal regulation is required to meet medical and psychological requirements or other prerequisites requiring this examination for the performance of the work. In other cases, an employer is not prohibited from asking an applicant to submit to a medical examination; however, this information must not be used in any way that discriminates against the applicant. Pursuant to the Act on Protection, Aid and Support of Public Health (Section 30, paragraph 14) [69], if an employer has reasonable doubts regarding an employee’s ability to perform assigned work tasks owing to a medical condition, the employer can, after agreement with the employee’s representative and physician, require the employee to undergo a medical exam related to the work performed. The employee is obliged to undergo this medical examination. The Labor Code [36] does not recognize any restrictions against alcohol and drug testing of applicants. It is the right of an employer to refuse to hire an applicant who refuses to submit to this test.

#### 3.1.21. Slovenia 

Due to the heightened risk of accidents at work, workers are prohibited from working or being at the workplace under the influence of alcohol, drugs, or other prohibited substances in accordance with the procedures and modalities defined by the employer in the safety statement and risk assessment document. The employer must suspend work and remove from the workplace any worker who is violating safety regulations. A fine anywhere in the range of EUR 100 to 1000 shall be imposed on a worker, who works or is at the workplace under the influence of alcohol, drugs, or other prohibited substances [70].

#### 3.1.22. Spain

Guidelines on the management of alcohol in the workplace are regulated in the Workers’ Statute, in detail in Article [45]. The employment contract can be revoked according to the decision of the employer through dismissal for a serious and culpable breach by the employee. Habitual drunkenness or alcohol addiction is considered a breach of contract due to the negative impact on work [71].

#### 3.1.23. Sweden

The Occupational Health and Safety Act states that the employer must “clarify which internal rules and procedures apply if an employee appears to be under the influence of alcohol or other intoxicants at work”. Employees have the right to know the workplace rules in regard to alcohol and drugs. In order to ensure clarity, the employer is obliged to create an alcohol and drug policy and an action plan. The policy should set out clear positions on how the workplace views alcohol and drugs, e.g., whether alcohol can be served at staff parties and what happens if someone arrives at work under the influence of alcohol. Business owners or managers can also turn to Alna for advice, support, and training on alcohol and drug problems in the workplace [72]. Currently, there are many proposals for action to regulate the management of alcohol and drugs in the workplace being evaluated and many are being approved. Proposals for action mainly pertain to increasing research into the relationship between alcohol and sick leave and focusing efforts on training for managers, safety representatives, and trade union representatives. The establishment of a register of alcohol-related workplace injuries and occupational illnesses has been indicated, as these are two of the most serious issues plaguing the Swedish economy. Both politicians and governing bodies and national alcohol awareness committees have implemented initiatives to strengthen action against alcohol and drug abuse at work. Moreover, information and discussion material on alcohol and drugs in the workplace have been developed to raise awareness of the issue among the working population [73].

### 3.2. Comparison of Alcohol Rules in the 27 Countries of the European Union

The analysis shows that there are three different approaches being undertaken by the member states of the union, one is to have laws passed regulating the phenomenon, that impose restrictions on the consumption of alcohol; another approach is to delegate to the employer the power to control the phenomenon (Ireland, Luxembourg, Portugal, and Sweden); and finally, in four countries there is no specific rule—according to our research—on the subject (Greece, Malta, Netherlands, and Romania) (Table 1).

Table 2 shows that there are different approaches to alcohol consumption: In some countries, it is forbidden to be intoxicated in the workplace, while in others, it is forbidden to drink any amount of alcohol. The monitoring of compliance with rules takes place through two possible methods: The first involves random tests and the second indicates discretionary tests. In both cases, the worker can refuse the test, but the consequences of the refusal are not always clear; there is also the question of consent that in many countries is a mandatory requirement; therefore, in case of refusal of consent, it is not possible to perform a test. In this regard, the interventions to be implemented in the event of non-compliance with the national rules, vary according to the country, and only in 11 countries we have found information regarding sanctions.

In almost all countries where alcohol control legislation has been found, there has been a specific indication that alcohol monitoring is mandatory only for certain occupational groups.

There are two different approaches finalized in monitoring the issue of alcohol in the workplace, the first involves a medical evaluation and the second the alcohol breath test (BAC).

Only in 10 countries (Table 2) a medical examination is required, and there are no regulatory references on the types of examination to be conducted or on minimum standards to be followed (such as biological matrix tests, psychological tests, clinical-functional evaluations, or other).

The comparative analysis shows that in all the countries that were examined there is an indication on the monitoring of alcohol intake through BAC control at least for professional drivers (Table 3). However, there is considerable variation in the limits imposed by local laws. The approach of some countries is that of a total ban on the intake of alcohol during the work shift, thus setting a limit of 0.00. In other countries, the limit is set at values ranging from 0.01 to 0.05; therefore, it is possible to consume alcohol in these countries but only to a moderate extent. 

## 4. Discussion

The issue of alcohol consumption in relation to work is a very complex issue in regard to two main aspects. On the one hand, the monitoring of alcohol use at work aims to verify compliance by the worker with the rules. The alternative aspect is to keep the focus purely on the issue of the health of the worker and therefore on the monitoring of the worker’s health conditions; the resulting process must allow for the implementation of preventative measures to reduce accidents at work and could affect other occupational exposures; for instance, excessive alcohol consumption could accelerate or slow down the metabolism of professional xenobiotics thus modulating the toxic effects of chemical agents [74].

The monitoring of alcohol intake in workers is also aimed at protecting the health of the general population. Prohibitions on alcohol intake exist, where present, only for certain professional categories that are employed in activities that may involve the general population (such as drivers, law enforcement, and doctors) [75,76,77,78].

The concept of liability to the general population also raises the issue of civil and criminal liability in the event of accidents caused by workers who work in breach of the rules and legislation on the use of alcohol. In countries where the person responsible for checking the health of the worker and the intake of alcoholic substances is the doctor, the responsibility for any damage to property and/or persons may be imputed to him (Italian law decree 81/2008) [57]. 

The significant degree of difference and variability in conduct between all the countries of the European Union, the different approaches to prescribing analysis (BAC or medical surveillance), the rules and legislations that require the need for monitoring to be carried out by a doctor or other professionals (which is possible with the examination of the BAC) creates a great difference in the responsibility of the occupational physician; this difference appears peculiar to the question of alcohol. Amantea et al. [11], for example, have pointed out that on the issue of vaccinations and doctors’ responsibilities in the 27 countries of the European Community there are very similar criminal and civil responsibilities; therefore, in the detection of this homogeneity, it appears that it would also be desirable to have a similar condition with regard to the obligations, duties, and responsibilities of the occupational physician in relation to checks for alcohol-related conditions. Furthermore, in a study on the medical liability of vaccinations conducted by Beccia et al. [12], it was noted that medical liability is mainly based on Civil Law, since in their study seven out of eighteen countries were based on Common Law; a more uniform behavior in the European Community should be sought for the medical liability of alcohol controls, as well.

As the comparative analysis has shown, there is a difference in approach between countries in relation to the imposition of sanctions in the event of transgression of workplace alcohol rules and regulations, the possibility of a refusal of the worker to be tested, and the obligation, or even the simple possibility, of the ability to conduct alcohol tests in the workplace. The subjects responsible for monitoring compliance with the rules vary from country to country, which indicates a significant difference between the various members of the European Community. Compliance with the rules in some cases is managed by the same companies (employer), while in other cases, such as in Italy, for example, there are controls carried out by subjects responsible for supervisory tasks on the laws related to safety in the workplace who, in case of non-compliance, can directly raise criminal and administrative sanctions.

In Italy, the doctor has a central role in all phases of evaluation and management of occupational risks and therefore also of the aspects that affect the safety and health of workers; being on duty under the influence of alcohol is certainly one of these conditions. The occupational physician through health surveillance in Italy is always directly involved; however, the presence of the doctor in Europe at large, for the purpose of alcohol control, is not a central figure and often the screening activity is delegated to other professionals (alcohol breath tests in some cases can also be performed by non-medical personnel). According to our analysis, tests must be performed by specialized health professionals in Latvia; but Italy seems to be the only country where the legislation, in addition to transferring the burden of controlling alcohol intake to employees, with an explicit mandate requires the occupational physician to also detect the absence of alcohol dependence in all workers who carry out activities in which they may be responsible for events that may impact the general population. This task appears to be incredibly complex given the difficulty of detecting addiction through simple screening tests and in a therapeutic setting not suitable for the analysis of this problem.

A further common problem in regard to the management of workers who have problems in relation to alcohol is the collection of the consent of workers for the execution of the tests. This occurs due to the national legislation of many countries that prevents these tests from being conducted in the absence of consent and therefore where there exists the possibility for the worker to avoid this examination, the local regulations then manage the denial of the worker in various ways through the application of measures in some cases.

Differing approaches were found among the different EU countries regarding alcohol testing. In Lithuania and Italy, the permitted blood alcohol concentration limits (BAC) for all types of workers are 0.02% and 0.00%, respectively. These are more restrictive policies. Similar values are found in other countries only in relation to professional drivers. The regulations in these two countries refer to a complete prohibition on the consumption of alcoholic beverages and the investigation of possible alcoholic consumption by employees at work, rather than identifying a condition of alcohol dependence. Moreover, it was revealed that employers in countries, such as the Czech Republic, Hungary, Lithuania, and Poland, have the right to conduct a test without evidence of clear intoxication. Again, this is a significantly stricter testing regime than in other countries.

An aspect shared by all European countries is that alcohol controls are conducted by specifically selecting the professional categories to which the monitoring programs should be addressed, but there are also profound differences in this area, as the categories vary widely from country to country. These profound differences between the precautions that are to be imposed on workers who drink alcoholic beverages inevitably entail that the circulation of professionals between the different countries becomes more complex.

## 5. Conclusions

This legislative revision in the European context reveals a strong lack of homogeneity among the 27 countries of the Union on the issue of the management of alcohol-related problems among workers. The regulations of the individual states on health and safety in the workplace are drawn up on the basis of community regulations; therefore, this lack of homogeneity appears to be misplaced and misguided. A commonly held view of the issue is also necessary since the movement of workers between European countries would be eased if there was agreement in rules and regulations across countries within the union. Otherwise, a worker could find themselves in a condition of being unable to work across a border since the blood alcohol level is unexpectedly different; this condition of non-homogeneity could lead the worker to be unable to practice in a neighboring country until they have carried out specific clinical examinations.

The issue of controls related to alcohol intake can also have an impact on the rights of the citizen and on the right to be able to refuse the test; therefore, it seems peculiar that in Europe different legal conditions can be created between countries. Moreover, a common vision of the problem could lead the scientific community to study systems for the detection of conditions that place workers and the community at risk (establishing standard core exams). 

Finally, the theme is of particular interest, since the problem of alcohol abuse is quite widespread in the population; the European Transport Safety Council claims on its website “We’re helping EU Member States share best practice on a wide range of road safety policies”; and going beyond the topic of road safety and extending the concept to occupational safety our research group argues that a task of homogenization with the identification of minimum levels of caution to be shared at the community level would be appropriate for workers. 

## Figures and Tables

**Table 1 ijerph-19-16964-t001:** Twenty-seven European countries and alcohol regulations.

Country	National Legislation in the Workplace	Medical Examination	Alcohol Test	Sanction	Require Testing	Allow Testing	BAC%
Austria	Yes	Yes	Yes	Yes	No	Yes	N.D.
Belgium	Yes	Yes	Yes	N.D.	No	Yes	N.D.
Bulgaria	Yes	N.D.	Yes	Yes	Yes	N.D.	N.D.
Croatia	Yes	Yes	Yes	N.D.	No	Yes	N.D.
Cyprus	Yes	Yes	Yes	Yes	No	Yes	N.D.
Czech Republic	Yes	Yes	Yes	Yes	Yes	No	N.D.
Denmark	Yes	N.D.	Yes	N.D.	No	Yes	N.D.
Estonia	Yes	N.D.	Yes	Yes	No	Yes	N.D.
Finland	Yes	Yes	Yes	N.D.	No	Yes	N.D.
France	Yes	N.D.	Yes	Yes	No	Yes	N.D.
Germany	Yes	N.D.	Yes	Yes	No	Yes	N.D.
Greece	N.D.	N.D.	N.D.	N.D.	N.D.	N.D.	N.D.
Hungary	Yes	N.D.	Yes	N.D.	Yes	No	N.D.
Ireland	Yes *	N.D.	Yes	N.D.	No	Yes	N.D.
Italy	Yes	Yes	Yes	Yes	No	Yes	0.00
Latvia	Yes	Yes	Yes	Yes	No	Yes	N.D.
Lithuania	Yes	Yes	Yes	Yes	No	Yes	0.02
Luxembourg	Yes *	Yes	Yes	N.D.	N.D.	Yes	N.D.
Malta	N.D.	N.D.	N.D.	N.D.	N.D.	N.D.	N.D.
Netherlands	N.D.	N.D.	N.D.	N.D.	N.D.	N.D.	N.D.
Poland	Yes	N.D.	Yes	Yes	Yes	No	N.D.
Portugal	Yes *	Yes	Yes	N.D.	No	Yes	N.D.
Romania	N.D.	N.D.	N.D.	N.D.	N.D.	N.D.	N.D.
Slovak Republic	Yes	Yes	Yes	N.D.	No	Yes	N.D.
Slovenia	Yes	N.D.	N.D.	Yes	N.D.	N.D.	N.D.
Spain	Yes	N.D.	N.D.	Yes	N.D.	N.D.	N.D.
Sweden	Yes *	N.D.	Yes	N.D.	N.D.	Yes	N.D.

* Company’s internal policies, N.D.: Not detected in literature review; BAC%: Blood alcohol concentration limit for workers (ETSC: European Transport Safety Council).

**Table 2 ijerph-19-16964-t002:** European countries and alcohol rules in the workplace (including companies’ internal policies).

Country	Rules Prohibit to Be Drunk	Rules Prohibit to Drink Any Alcohol	Random Test	Discretionary Test	Worker Can Refuse Test	Different Rules for Different Occupations
Austria	Yes	Yes	N.D.	Yes	Yes	Yes
Belgium	Yes	Yes	N.D.	Yes	Yes	Yes
Bulgaria	Yes	Yes	N.D.	Yes *	N.D.	Yes
Croatia	Yes	N.D.	N.D.	Yes	Yes	Yes
Cyprus	Yes	Yes	N.D.	Yes	N.D.	N.D.
Czech Republic	Yes	Yes **	Yes	Yes	No	Yes
Denmark	Yes	N.D.	N.D.	Yes	Yes	N.D.
Estonia	Yes	N.D.	N.D.	Yes	Yes	N.D.
Finland	Yes	N.D.	N.D.	Yes	Yes	N.D.
France	Yes	Yes ***	No	Yes *	N.D.	Yes
Germany	Yes *	Yes *	N.D.	Yes *	N.D.	Yes
Greece	N.D.	N.D.	N.D.	N.D.	N.D.	N.D.
Hungary	Yes *	Yes *	N.D.	Yes *	No *	Yes
Ireland	Yes	Yes	No	Yes	Yes	N.D.
Italy	Yes	Yes	Yes	Yes	No	Yes
Latvia	Yes	N.D.	N.D.	Yes	N.D.	N.D.
Lithuania	Yes	Yes	N.D.	Yes	N.D.	Yes
Luxembourg	Yes	Yes	N.D.	Yes	N.D.	N.D.
Malta	N.D.	N.D.	N.D.	N.D.	N.D.	N.D.
Netherlands	N.D.	N.D.	N.D.	N.D.	N.D.	N.D.
Poland	Yes	Yes	Yes	N.D.	No	N.D.
Portugal	Yes	Yes	Yes *	Yes	N.D.	Yes
Romania	N.D.	N.D.	N.D.	N.D.	N.D.	N.D.
Slovak Republic	Yes	Yes	N.D.	Yes	N.D.	Yes
Slovenia	Yes	N.D.	N.D.	N.D.	N.D.	N.D.
Spain	Yes	N.D.	N.D.	N.D.	N.D.	N.D.
Sweden	Yes	Yes	Yes	Yes	Yes	N.D.

* Only for specific categories of workers. ** Exception: Employees who work in adverse microclimate conditions can consume beverage with reduced alcohol content. *** Except for wine, beer, cider, or perry; may be served in the company restaurant or in special.

**Table 3 ijerph-19-16964-t003:** Blood alcohol concentration limit for commercial drivers.

Country	BAC% Commercial Drivers
Austria	0.01
Belgium	0.02
Bulgaria	0.05
Croatia	0.00
Cyprus	0.02
Czech Republic	0.00
Denmark	0.05
Estonia	0.00
Finland	0.05
France	0.05
Germany	0.00
Greece	0.02
Hungary	0.00
Ireland	0.02
Italy	0.00
Latvia	0.05
Lithuania	0.00
Luxembourg	0.02
Malta	0.02
Netherlands	0.05
Poland	0.02
Portugal	0.02
Romania	0.00
Slovak Republic	0.00
Slovenia	0.00
Spain	0.03
Sweden	0.02

BAC% commercial drivers: Blood alcohol concentration limit for commercial drivers (ETSC: European Transport Safety Council).

## Data Availability

Not applicable.

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
