# Peer review of "Alcohol Consumption in the Workplace: A Comparison between European Union Countries’ Policies"

_ijerph, 2022, doi:10.3390/ijerph192416964_

Round 1

Reviewer 1 Report

This paper presents all the European Union countries’ policies around alcohol consumption in workplace. A lot of work has gone into the results section of the study. The paper would benefit from major revision. I have highlighted some main points below and have added some more detailed comments in the main paper.

Overall, this paper seems to be in a draft stage where it needs to be carefully reviewed. As a comparison study the results section needs to be improved, the discussion needs to be strengthened and the paragraphs/ sentences need to be carefully revised. There are many long sentences with more than one point that have made the main concept written very difficult to understand.

Detailed review

Starting from Line 83, the method section, needs some revision on the flouncy of the steps performed (e.g. peer reviewed literature and grey literature search). I was wondering why the authors have chosen only PubMed as a scientific database for the peer reviewed literature? In addition, I suggest if the authors add the number of the links they have considered when searching google ( Line 97) (e.g. the first 10 or 20 links shown as a result of each google search).

According to table 1 (Line 501), there are 8 countries that their alcohol legislations were not detected from this literature review! While in the results (Line 107), the authors state that only four countries have no legislations around alcohol consumption in working place. According to the text, I understand that the other four countries (Ireland, Luxembourg, Portugal and Sweden) have some other type of legislations such as voluntary guidelines for testing and case by case management. Therefore, these details need to be more clear in the summary of the results and the table. You might consider adding other than N.D. abbreviation or additional columns to clarify that.

I do appreciate all the work that has been placed into the result section. However, this part has presented each country as a separate case study. I believe this work needs to be further analysed. For the purpose of the topic of the manuscript “comparison”, I suggest that the authors present these results (maybe followed with analysis) as a cohesive work where a comparison between countries are presented. This can be achieved by adding a detailed table of the existing policies. Maybe by having a global recommendations framework as a reference and/ or choosing few common/ or unique challenges or themes to stitch the countries together.

The discussion starts from Line 504, where I have added more detailed comments within the text. Overall, because of the points mentioned below the discussion needs major revision.

-          As a first paragraph in the discussion, this paragraph was expected to focus on summarising the key findings in a concise language. Unfortunately, it starts with long and complex sentences which do not convey a main idea.  

-          There are few sentences (Lines 516-531) introducing what has been found in the previous two comparison studies done by the same group on COVID-related policies and stating that these are interesting to investigate ...... If these sentences are meant to be as recommendations for further studies, then they are better to be mentioned at the end. Otherwise, the purpose of these few sentences here is unclear!

-          Along the discussion, there are only three references which repeatedly come at the end (Lines 604, 610 and 614). I believe that the authors need more work to place this study within the context of previous studies.

-          There are very long paragraphs with extremely long sentences that have made the whole discussion really difficult to understand (e.g. Lines 580 to 625).

-          I suggest that the authors consider adding some analysis to the results where they can use their results to clarify what they are comparing (the alcohol consumption in workplace policies between the 27 countries). These include the agenda setting, problem definition, policy development and policy implementation. Then the authors can use the discussion to highlight what was unexpected (e.g. some European countries have no legislation in place or e.g. some European countries differ in their approaches).

This paper has used a lot of grey literature, where I believe that their manual referencing needs to be improved to contain more information than only website and retrieved day. For example, lines 653, 655 and 719, the references are missing many details such as author name and published day. I also appreciate if the authors have a translation in brackets for the non-English headings.

Author Response

Rome, November 28, 2022

Dear Reviewer,

please find attached the revision of our manuscript entitled “Alcohol consumption in the workplace: a comparison between European Union countries’ policies”prepared by Ivan Borrelli, Paolo Emilio Santoro, Maria Rosaria Gualano, Antongiulio Perrotta, Alessandra Daniele, Carlotta Amantea and Umberto Moscato. We appreciate the time and effort that you allocated to evaluate the manuscript critically and objectively to improve the scientific quality of our study. Authors’ point-by-point comments are listed below and all the revised parts in the manuscript as well as our responses to the reviewer are marked blue.

We hope this revised paper is suitable for publication in International Journal of Environmental Research and Public Health and we look forward to your consideration.

Sincerely yours,

Ivan Borrelli

Reviewer 1

This paper presents all the European Union countries’ policies around alcohol consumption in workplace. A lot of work has gone into the results section of the study. The paper would benefit from major revision. I have highlighted some main points below and have added some more detailed comments in the main paper.

Overall, this paper seems to be in a draft stage where it needs to be carefully reviewed. As a comparison study the results section needs to be improved, the discussion needs to be strengthened and the paragraphs/ sentences need to be carefully revised. There are many long sentences with more than one point that have made the main concept written very difficult to understand.

Thank you very much for agreeing to review this research. We have found your comments enlightening and have provided to make the necessary changes. 

  1. Starting from Line 83, the method section, needs some revision on the flouncy of the steps performed (e.g. peer reviewed literature and grey literature search). I was wondering why the authors have chosen only PubMed as a scientific database for the peer reviewed literature? In addition, I suggest if the authors add the number of the links they have considered when searching google (Line 97) (e.g. the first 10 or 20 links shown as a result of each google search).

Answer:

Specifically, a legislative review was conducted on all 27 EU Member States to screen legislation on alcohol in the workplace, through a ''grey literature'' search, using the databases of European and Member States' institutional repositories and the Google search engine. The search was then extended to the scientific literature, using the PubMed database, when no specific regulations emerged from the databases of governmental sites. About the links consulted on the Google search engine, when results inherent to the aim of the review emerged, the first 10 links shown were the ones that were followed, while the search continued up to 50 links, when no legislation or pertinent material was found. If the reviewer considers it appropriate, can we include this in the methods.

The intention of the search is of legislative type and we have not considered it useful to proceed with the PRISMA methods of systematic review.

Action:

In thanking the reviewer for the comment, we have we reworked the text; follows the new elaborate (lines 94-106):

A legislative review was carried out on all 27 EU Member States. A narrative review methodology was adopted to screen workplace alcohol legislation through a “grey literature” search using the databases of European and Member State institutional repositories and the Google search engine. The search was then extended to the scientific literature, using the PubMed database, when no specific regulations emerged from the databases of governmental sites. To identify the national legislation in force in each country, the national archives in the original language were screened (labor code, civil code laws related to workplace safety, occupational health and safety laws and so on), to highlight the policies on alcohol use in the workplace related to consumption, testing or consequences.

The scientific literature research on PubMed – when required – was conducted selecting studies published over a 10-year period (January 2012 - June 2022); for each EU country, studies mentioning alcohol and drug use in the workplace were included and screened for relevant legislation mentioned in the paper.

  1. According to table 1 (Line 501), there are 8 countries that their alcohol legislations were not detected from this literature review! While in the results (Line 107), the authors state that only four countries have no legislations around alcohol consumption in working place. According to the text, I understand that the other four countries (Ireland, Luxembourg, Portugal and Sweden) have some other type of legislations such as voluntary guidelines for testing and case by case management. Therefore, these details need to be more clear in the summary of the results and the table. You might consider adding other than N.D. abbreviation or additional columns to clarify that.

Answer:

We thank the reviewer for his comment that allows us to make the text clearer.

We made it clear in the text that for 4 countries no reference was found on the research subject (Greece, Malta, Netherlands, Romania), while for another four there is no explicit provision in the Labour Code covering the management of alcohol in the workplace (Ireland, Luxembourg, Portugal, or Sweden) but other indication for employee or employer are provided.

Action:

In order to make the text clearer, we have changed the term "N.D." (No Data) in table 1 with "company's internal policies"; we explained in the new section the data (3.2 Comparison) (lines 519-526):

The analysis carried out shows that there are three different approaches by the member states of the union, one is to have passed laws regulating the phenomenon that impose restrictions on the consumption of alcohol; another approach is to delegate to the employer the power to control the phenomenon (Ireland, Luxembourg, Portugal, and Sweden); and finally, in 4 countries there is no specific rule – in our research – on the subject (Grece, Malta, Netherlands, Romania) (Tab.1).

  1. I do appreciate all the work that has been placed into the result section. However, this part has presented each country as a separate case study. I believe this work needs to be further analysed. For the purpose of the topic of the manuscript “comparison”, I suggest that the authors present these results (maybe followed with analysis) as a cohesive work where a comparison between countries are presented. This can be achieved by adding a detailed table of the existing policies. Maybe by having a global recommendations framework as a reference and/ or choosing few common/ or unique challenges or themes to stitch the countries together.

Answer:

We thank the reviewer for his valuable suggestion.

Action:

We reorganized the results in two section, in the first we thought it useful to leave the description work separated by country. In the second section (3.2) we have highlighted the differences between countries  with new comparative tables (lines 519-607).

  1. The discussion starts from Line 504, where I have added more detailed comments within the text. Overall, because of the points mentioned below the discussion needs major revision.

Answer:

Thank you, we followed the recommendation and modified the discussion.

Action:

To improve the comprehension and English of the paragraph we have reworked the whole text. (lines 609-690):

  1. As a first paragraph in the discussion, this paragraph was expected to focus on summarising the key findings in a concise language. Unfortunately, it starts with long and complex sentences which do not convey a main idea.

Answer:

Thank you

Action:

We have rephrased (lines 612-621):

The other aspect is to keep the focus purely on the health issue and therefore on the monitoring of health conditions of the worker; the final process must allow the implementation of preventive measures to reduce accidents at work and the could affect other occupational exposures; excessive alcohol consumption could accelerate or slow down the metabolism of professional xenobiotics thus modulating the toxic effects of chemical agents[75].

The monitoring of alcohol intake in workers is also aimed at protecting the health of the general population, in fact the prohibitions on alcohol intake exist (where present) only for certain professional categories that are employed in activities that may involve the general population (such as drivers, law enforcement, doctors). [76–79l.

  1. There are few sentences (Lines 516-531) introducing what has been found in the previous two comparison studies done by the same group on COVID-related policies and stating that these are interesting to investigate ...... If these sentences are meant to be as recommendations for further studies, then they are better to be mentioned at the end. Otherwise, the purpose of these few sentences here is unclear!

Thank you, we have rephrased (lines 622-642):

The concept of liability to the general population also raises the issue of civil and criminal liability in the event of accidents caused by workers who work in breach of the rules on the use of alcohol. In countries where the person responsible for checking the health of the worker and the intake of alcoholic substances is the doctor, responsibility for any damage to property and/or persons may be imputed to him (Italian law decree 81/2008).

The great difference in behavior between different countries, the different approaches to prescribing analysis (BAC or medical surveillance), the reminder by rule of the need for monitoring to be carried out by the doctor or other professionals (which is possible with the examination of the BAC) creates a great difference in responsibility of the occupational physician; this difference appears peculiar to the question of alcohol,  Amantea et al. [11] for example, have pointed out that on the issue of vaccinations and doctors responsibilities in the 27 countries of the European Community they have a very similar criminal and civil responsibilities; thus detecting a homogeneity, it would be desirable to have an equal condition also with regard to the obligations, duties and responsibilities of the occupational physician in relation to checks for alcohol-related conditions. Furthermore, in a study on the medical liability of vaccinations conducted by Beccia et al. [12], it was noted that medical liability is mainly based on Civil Law since in their study seven out of 18 countries were based on Common Law; a more uniform behavior in the European Community should be sought for the medical liability of alcohol controls, too.

  1. Along the discussion, there are only three references which repeatedly come at the end (Lines 604, 610 and 614). I believe that the authors need more work to place this study within the context of previous studies.

Answer

We thank the reviewer for the advice.

Action:

We add new bibliography (now in the text you will find the following articles: 11-12-58-75-76-77-78-79.

  1. There are very long paragraphs with extremely long sentences that have made the whole discussion really difficult to understand (e.g. Lines 580 to 625).

Answer

Thanking the reviewer we have preferred delete the part about the MRO to make the discussion more focused on the primary topic

Action:

The part was removed

  1. I suggest that the authors consider adding some analysis to the results where they can use their results to clarify what they are comparing (the alcohol consumption in workplace policies between the 27 countries). These include the agenda setting, problem definition, policy development and policy implementation. Then the authors can use the discussion to highlight what was unexpected (e.g., some European countries have no legislation in place or e.g. some European countries differ in their approaches).

Answer:

Thank you for the suggestion proposed.

Action:

We have added table 2 and and table 3; we have implemented table 1 and discussed it in new paragraph (3.2 Comparison of alcohol rules in the 27 countries of the European Union); all table are in the paragraph 3.2 (line 519). We also implemented the discussion.

  1. This paper has used a lot of grey literature, where I believe that their manual referencing needs to be improved to contain more information than only website and retrieved day. For example, lines 653, 655 and 719, the references are missing many details such as author name and published day. I also appreciate if the authors have a translation in brackets for the non-English headings.

Answer:

Thanking the reviewer for the comment, we have adjusted the bibliography by putting the English translation in brackets for non-English bibliographic references. We improved also the reference with only website.

Reviewer 2 Report

This paper presents useful and interesting information on national policies around alcohol in the workplace. The authors should be commended on having retrieved this data, and it would be good to see this information in the public domain. However, the manuscript needs significant revision before it is ready for publication.

In the introduction, the discussion of alcohol harm, and the prevalence of alcohol problems, needs to be much more precise and accurate. For example, it is stated that ‘alcohol is an ever-growing problem in the population’. In fact, alcohol consumption has been falling, especially among younger people, for a number of years (though alcohol harms remain significant among older people), and this is especially true in Europe. Critically, trends also vary considerably by country, including the 27 considered here. The authors should ensure their core prevalence and harm data is up to date (the WHO Global Status Report is useful), and they should avoid generalised statements that imply drinking cultures or behaviours are the same in the different settings.

Terms describing the potential effects of intoxication (e.g the ‘psychological integrity of the worker’) are not well defined. There tends to be an assumption the any drinking in the workplace is liable to produce very harmful consequences, whereas there is clearly a continuum of risk. A worker breaching the 0.02% BAC limit applied in Latvia, for example, may well not be meaningfully impaired (in fact, may not even have consumed alcohol on the day in question), and may not be facing a health risk, whereas workers drinking at higher levels will present more serious risks. This should be reflected in the discussion as it has ethical implications regarding the rights of workers.

The discussion of dependency needs more nuance and precision. The authors seem to suggest that testing for dependence could, and perhaps should, form part of workplace screening – but this is a very controversial proposal, given the complexity of defining and diagnosing alcohol dependence. I think the authors should avoid taking a position on this issue, and simply report whether it takes place. Alternatively, if they do wish to take a position on the pros and cons of such an approach, they will need to engage more deeply with the extensive literature on alcohol dependency, and the problems around how it is defined and diagnosed. Terms such as ‘morbidly dependent’ (line 142) should be used with caution.

I would suggest that the country-by-country narrative sections are further divided into subsections to help the reader compare between them. At the moment it is a little hard to follow. It is up to the authors, but it seems to me that some of the core considerations, which could form subheadings, are:

·      Do the rules prohibit being drunk in the workplace, or include drinking any alcohol in the workplace? These are quite different.

·      Are tests random or discretionary? Random testing has significantly different ethical implications to testing when a worker appears obviously intoxicated

·      Can workers refuse tests?

·      Are there different rules for different occupations?

·      Are specific blood alcohol limits applied? If so, what are they?

On line 35 it says ‘There is no legislation in Ireland that requires employers to test their employees for alcohol or drugs in the workplace’. This raises an important question that needs to be discussed more clearly in the paper: does the legislation in the various countries  *require* testing or does it *allow* testing. These are two very different legal conditions, with very different implications for practice.

The section on Latvia suggests that a BAC over 0.02% is prohibited under all circumstances, and that employers have the right to test this without evidence of obvious intoxication (which would be very unlikely in the case of someone with a BAC of, say, 0.05%). Again, this is a much more severe testing approach than other countries, so it would be good to draw out the ethical implications of this, as well as other strict codes (e.g. Poland’s policy of allowing immediate dismissal for possessing alcohol at work).

Lines 446-7 The comment about industry influence isn’t clear here. Other countries with strict policies (e.g. France) also have high consumption and influential producers, so it isn’t clear why Portugal would be particularly affected by this.

The wording on lines 454-7 suggests the Slovak Republic has a policy of regular random workplace testing, is that the case? Or is it that employers are allowed (but not required) to do this under law if they wish (as above, requiring such testing and simply allowing it are very different).

Table 1.27: I suggest this table is reconsidered to include some of the key components discussed above (e.g. is random testing allowed / required). That may allow for fewer ND findings than is the case currently, where the focus is type of testing, and for better comparison of policies.

In Table 1.27, it would be useful to distinguish between ‘ND’ (no data has been detected) and ‘No policy’ (i.e. available data shows there is no policy in place). E.g. in the narrative section on Portugal, it is stated that there is no legislation on alcohol in the workplace, but here it is ambivalent because it is reported as ‘no data detected’.

The discussion section suggests that health is a primary reason, and concern for, workplace testing but it is not clear why this is assumed. Employers may have a concern for employee health, but workplace drug and alcohol testing is more likely to be focused on risk (of injury, to reputation, to productivity etc.). Here risks and responsibility are presented as secondary considerations, but it isn’t clear why. I think the discussion of motivations for testing needs more work in order to be more compelling.

Lines 517-8 – the point about criminal liability is well made, but would it not also be interesting to consider the wider human rights implications for workers when subject to random testing?

A number of sections in the conclusion repeat large sections of text. This should have been noticed in proofreading prior to submission. It is frustrating to deal with this as a reviewer.

Lines 531-3: I don’t understand this section on supervision, so it may need rephrasing.

Line 541-50 is a very long, multi-clause sentence which covers a number of themes. It is very hard to follow. The same is true for lines 553-60. The paper needs careful editing to iron out such overly-complicated sentences.

Lines 567-8 strongly suggest that education and training are the most effective forms of prevention when it comes to alcohol. The evidence on prevention suggests otherwise: that education and training – unless done very well – can, in fact, be ineffective. I suggest this is either removed, or the authors familiarise themselves with more of the evidence on prevention.

Lines 580-600 are hard to follow. As mentioned above, the discussion of dependence needs to be more precise (what does it mean to ‘shift into a state of real dependence’, for example?). Is the proposal to require MROs in all workplaces realistic? How much might it cost? What evidence is there to demonstrate MROs are effective in achieving these wider goals (beyond simply verifying the accuracy of test results)? Any evidence should be presented here, since the authors appear to be recommending this approach.

Lines 610-25 are also repeated material.

The conclusion suggests that a lack of homogeneity is a bad thing, but without explaining why. Is homogeneity necessary given the considerable variation in drinking profiles, prevalence and harms across the 27 countries? Importantly, are the authors proposing the standard approach should be strict (e.g. Latvia) or less intrusive. The arguments in this section need to be made much more carefully.

Line 630: I suggest removing the word ‘endemic’ as it is poorly-defined and doesn’t reflect the heterogeneity of drinking cultures across, and within, countries.

I think the conclusion needs to return to key issues around e.g. implementation (what might the practical implications of adopting workplace testing be), human rights (what are the implications of random testing or summary dismissal for the rights of workers), levels of severity across different regimes, and the difference between banning alcohol from the workplace and testing for drunkenness. At the moment it only briefly sets out the case for a more standardised approach.

For these reasons, while the basic research is valuable, I think extensive contextual work is needed before it can be considered for publication.

Author Response

Rome, November 28, 2022

Dear Reviewers,

please find attached the revision of our manuscript entitled “Alcohol consumption in the workplace: a comparison between European Union countries’ policies”prepared by Ivan Borrelli, Paolo Emilio Santoro, Maria Rosaria Gualano, Antongiulio Perrotta, Alessandra Daniele, Carlotta Amantea and Umberto Moscato. We appreciate the time and effort that you allocated to evaluate the manuscript critically and objectively to improve the scientific quality of our study. Authors’ point-by-point comments are listed below and all the revised parts in the manuscript as well as our responses to the reviewer are marked blue.

We hope this revised paper is suitable for publication in International Journal of Environmental Research and Public Health and we look forward to your consideration.

Sincerely yours,

Ivan Borrelli

Reviewer 2

This paper presents useful and interesting information on national policies around alcohol in the workplace. The authors should be commended on having retrieved this data, and it would be good to see this information in the public domain. However, the manuscript needs significant revision before it is ready for publication.

Thank you very much for agreeing to review this research. We have found your comments enlightening and have provided to make the necessary changes. 

  1. In the introduction, the discussion of alcohol harm, and the prevalence of alcohol problems, needs to be much more precise and accurate. For example, it is stated that ‘alcohol is an ever-growing problem in the population’. In fact, alcohol consumption has been falling, especially among younger people, for a number of years (though alcohol harms remain significant among older people), and this is especially true in Europe. Critically, trends also vary considerably by country, including the 27 considered here. The authors should ensure their core prevalence and harm data is up to date (the WHO Global Status Report is useful), and they should avoid generalised statements that imply drinking cultures or behaviours are the same in the different settings.

Answer:

We thank the reviewer for giving us the chance to deepen the themes of the introduction.

Action:

We introduced new data (whit references) and we modified the introduction with the aim to be more accurate and precise (lines 35-57).

The use of alcohol while working is a significant phenomenon that constitutes an additional risk to pre-existing occupational hazards; it is estimated that between 5% and 20% of the working population in Europe have serious problems related to their alcohol usage [1].

Alcohol use is an important issue in the workplace because it can increase accidents, injuries, absenteeism, and inappropriate behavior by reducing a worker’s psycho-physical integrity, and significantly affecting the health and safety of third parties [2–5].

Workers under the influence of alcohol can in fact be a danger to others, and therefore a risk factor, if they use alcohol during work, especially in those occupational activities involve a high risk of accidents at work (e.g. police force, construction, transport, etc.) [6].

Although total alcohol consumption per capita in Europe fell from 12.3 liters in 2005 to 9.8 liters in 2016, [7] it remains among the highest in the world [8].

According to Eurostat statistics, in 2019, 8.4% of the adult working-age population in the EU consumed alcohol daily; 28.8% weekly; 22.8% monthly. Daily alcohol intake was most frequent in Portugal, with 20.7%, followed by Spain with 13.0%, and Italy with 12.1%. In the Netherlands, almost half the population, 47.3%, consumed alcohol weekly, followed by Luxembourg with 43.1% and Belgium with 40.8%. By comparison, monthly consumption in the EU was highest in Lithuania at 31.3%. In all EU Member States, men consume alcohol more frequently than women [9].

Alcohol-related harms are a severe public health problem, accounting for over 7% of illnesses or premature death in the EU. Even moderate alcohol consumption indeed increases the long-term risk of some heart disease, liver disease, and cancer, and frequent consumption of large amounts can be addictive. [4]

  1. Terms describing the potential effects of intoxication (e.g the ‘psychological integrity of the worker’) are not well defined. There tends to be an assumption the any drinking in the workplace is liable to produce very harmful consequences, whereas there is clearly a continuum of risk. A worker breaching the 0.02% BAC limit applied in Latvia, for example, may well not be meaningfully impaired (in fact, may not even have consumed alcohol on the day in question), and may not be facing a health risk, whereas workers drinking at higher levels will present more serious risks. This should be reflected in the discussion as it has ethical implications regarding the rights of workers.

Answer: Thank you for your comment; The main theme is the monitoring of compliance with labor standards and regulations; the limits are set, most of the time, to impose a ban on alcohol hiring and therefore the low limits of 0.00 or 0.02 do not serve to detect an intoxication but to detect a transgression of the rules.

Action:

We have inserted this concept in the new paragraph (3.2 comparison) and in the discussions.

  1. The discussion of dependency needs more nuance and precision. The authors seem to suggest that testing for dependence could, and perhaps should, form part of workplace screening – but this is a very controversial proposal, given the complexity of defining and diagnosing alcohol dependence. I think the authors should avoid taking a position on this issue, and simply report whether it takes place. Alternatively, if they do wish to take a position on the pros and cons of such an approach, they will need to engage more deeply with the extensive literature on alcohol dependency, and the problems around how it is defined and diagnosed. Terms such as ‘morbidly dependent’ (line 142) should be used with caution.

Answer:

Thanks for your comment. In Italy it is specified in the law that the occupational physician must exclude conditions of alcohol dependence during medical surveillance at least one time each year). It was our intention to put the question to the reader, in our opinion it is a mandate (to detect a dependency) that cannot be pursued in a screening visit; the treatment of addiction is a complex issue that in this discussion would take work off topic, also because in our analysis this mandate is specified only for Italy.

Action:

We have revised the text to address your concerns and decide to replace the term “morbidly dependent” with “employees suffering from alcohol addiction” to better let the reader understand that we are talking about an illness and employees need to be safeguarded (line 143).

  1. I would suggest that the country-by-country narrative sections are further divided into subsections to help the reader compare between them. At the moment it is a little hard to follow. It is up to the authors, but it seems to me that some of the core considerations, which could form subheadings, are:

  1. Do the rules prohibit being drunk in the workplace, or include drinking any alcohol in the workplace? These are quite different.
  2. Are tests random or discretionary? Random testing has significantly different ethical implications to testing when a worker appears obviously intoxicated
  3. Can workers refuse tests?
  4. Are there different rules for different occupations?
  5. Are specific blood alcohol limits applied? If so, what are they?:

Answer:

Thank you for the suggestion proposed. We have included the required data – if available - in the table 1 and table 2 (new table) and argued the results in paragraph 3.2.

Action

We added a new table (n.2) and improved table 1.

  1. On line 351 it says, ‘There is no legislation in Ireland that requires employers to test their employees for alcohol or drugs in the workplace’. This raises an important question that needs to be discussed more clearly in the paper: does the legislation in the various countries *require* testing or does it *allow* testing. These are two very different legal conditions, with very different implications for practice.

Answer:

Thank you for your valuable suggestion.

Action:

In order to discuss more clearly the issue of the countries  *require* testing or does it *allow* testing in the workplace, very different legal conditions, as the reviewer rightly points out, we have added four columns (Require testing; Allow testing; % BAC Commercial drivers; % BAC other workers) in table 1 on this issue.

  1. The section on Latvia suggests that a BAC over 0.02% is prohibited under all circumstances, and that employers have the right to test this without evidence of obvious intoxication (which would be very unlikely in the case of someone with a BAC of, say, 0.05%). Again, this is a much more severe testing approach than other countries, so it would be good to draw out the ethical implications of this, as well as other strict codes (e.g. Poland’s policy of allowing immediate dismissal for possessing alcohol at work).

Answer:

Thank you for your comment

Action:

in this regard we have included in table 1 of the results “27 European countries and Alcohol regulations” additional columns with the % of BAC in workers (line 528); we added a new table (tab. 3) with BAC limit in commercial drivers (line 572). We implemented the discussion with these themes lines 675-684).

A different approach was found among the different EU countries regarding alcohol test. In Lithuania and Italy, the permitted Blood Alcohol Concentration limits (BAC) for all types of workers are 0.02% and 0.00%, respectively. These are more restrictive policies. Similar values are found in other countries only in relation to professional drivers. The regulations in these countries refer to a complete prohibition on the consumption of alcoholic beverages and the investigation of possible alcohol consumption by employees at work, rather than identifying a condition of alcohol dependence. It was also revealed that employers in countries such as the Czech Republic, Hungary, Lithuania, and Poland have the right to conduct the test without evidence of obvious intoxication. Again, this is a much stricter testing regime than in other countries.

  1. Lines 446-7 The comment about industry influence isn’t clear here. Other countries with strict policies (e.g. France) also have high consumption and influential producers, so it isn’t clear why Portugal would be particularly affected by this.

Answer

We thank the reviewer for their input

Action:

we decided to remove the sentence (and the reference) since it was potentially misleading and irrelevant to the aim of the review.

  1. The wording on lines 454-7 suggests the Slovak Republic has a policy of regular random workplace testing, is that the case? Or is it that employers are allowed (but not required) to do this under law if they wish (as above, requiring such testing and simply allowing it are very different).

Answer

Thank you for pointing this out.

Action:

We decided to remove the sentence to rephrase the text (lines 469-482).

An employer is obliged to set a medical examination as a precondition of employment only if the applicant is a juvenile or for such jobs where a special legal regulation is required to meet medical and psychological requirements or other prerequisites requiring such an examination for the work to be carried out. In other cases, an employer is not prohibited from asking an applicant to submit to a medical examination; however, such information must not be used in any way that discriminates against the applicant. Pursuant to the Act on Protection, Aid and Support of Public Health (section 30 e, paragraph 14)[70], if an employer has reasonable doubts regarding an employee’s ability to perform assigned work tasks owing to a medical condition, the employer can, after agreement with the employee’s representative and physician, require the employee to undergo a medical exam related to the work performed. The employee is obliged to undergo such medical examination. The Labor Code [36] does not recognize any restrictions against alcohol and drug testing of applicants. It is the right of an employer to refuse to hire an applicant who refuses to submit to such a test.

  1. Table 1.27: I suggest this table is reconsidered to include some of the key components discussed above (e.g. is random testing allowed / required). That may allow for fewer ND findings than is the case currently, where the focus is type of testing, and for better comparison of policies.

Answer:

Thank you for the suggestion

Action:

We have remodulated in the table (line 528).

  1. In Table 1.27, it would be useful to distinguish between ‘ND’ (no data has been detected) and ‘No policy’ (i.e. available data shows there is no policy in place). E.g., in the narrative section on Portugal, it is stated that there is no legislation on alcohol in the workplace, but here it is ambivalent because it is reported as ‘no data detected’.

Answer:

Thank you for the suggestion

Action:

We have remodulated in the table (line 528).

  1. The discussion section suggests that health is a primary reason, and concern for, workplace testing but it is not clear why this is assumed. Employers may have a concern for employee health, but workplace drug and alcohol testing is more likely to be focused on risk (of injury, to reputation, to productivity etc.). Here risks and responsibility are presented as secondary considerations, but it isn’t clear why. I think the discussion of motivations for testing needs more work in order to be more compelling.

Answer:

Thank you for the suggestion

Action:

We have remodulated the discussion as follows (lines 610-621):

The issue of alcohol consumption in relation to work is a very complex issue with regards mainly to two aspects. On one hand, the monitoring activity on alcohol use at work aims to verify compliance - by the worker - with the rules. The other aspect is to keep the focus purely on the health issue and therefore on the monitoring of health conditions of the worker; the final process must allow the implementation of preventive measures to reduce accidents at work and the could affect other occupational exposures; excessive alcohol consumption could accelerate or slow down the metabolism of professional xenobiotics thus modulating the toxic effects of chemical agents[75].

The monitoring of alcohol intake in workers is also aimed at protecting the health of the general population, in fact the prohibitions on alcohol intake exist (where present) only for certain professional categories that are employed in activities that may involve the general population (such as drivers, law enforcement, doctors). [76–79l.

  1. Lines 517-8 – the point about criminal liability is well made, but would it not also be interesting to consider the wider human rights implications for workers when subject to random testing?

Answer:

Thank you for the comment, the part on human rights is of particular interest and importance; this aspect could bring this the discussion out of scope which aims to maintain a more focused view on the operators involved in monitoring and on the rules related to them.

Action:

We have remodulated the introduction of the discussion as follows (lines 622-653):

The concept of liability to the general population also raises the issue of civil and criminal liability in the event of accidents caused by workers who work in breach of the rules on the use of alcohol. In countries where the person responsible for checking the health of the worker and the intake of alcoholic substances is the doctor, responsibility for any damage to property and/or persons may be imputed to him (Italian law decree 81/2008) (58).

The great difference in behavior between different countries, the different approaches to prescribing analysis (BAC or medical surveillance), the reminder by rule of the need for monitoring to be carried out by the doctor or other professionals (which is possible with the examination of the BAC) creates a great difference in responsibility of the occupational physician; this difference appears peculiar to the question of alcohol,  Amantea et al. [11] for example, have pointed out that on the issue of vaccinations and doctors responsibilities in the 27 countries of the European Community they have a very similar criminal and civil responsibilities; thus detecting a homogeneity, it would be desirable to have an equal condition also with regard to the obligations, duties and responsibilities of the occupational physician in relation to checks for alcohol-related conditions. Furthermore, in a study on the medical liability of vaccinations conducted by Beccia et al. [12], it was noted that medical liability is mainly based on Civil Law since in their study seven out of 18 countries were based on Common Law; a more uniform behavior in the European Community should be sought for the medical liability of alcohol controls, too.

As the comparative analysis has shown, there is a difference in approach between countries in relation to the presence of sanctions in the event of alcohol rules' transgression, the possibility for the worker to refuse to be tested, the obligation, or the simple possibility, of carrying out the alcohol tests in the workplace. The subjects responsible for monitoring compliance with the rules are different from country to country, also showing in this aspect a difference between the various members of the European Community. In fact, compliance with the rules in some cases is managed by the same companies (employer) while in other cases such as in Italy, for example, there are controls carried out by subjects with supervisory tasks on the laws relating to safety in the workplace who, in case of non-compliance, can directly raise criminal and administrative sanctions.

  1. A number of sections in the conclusion repeat large sections of text. This should have been noticed in proofreading prior to submission. It is frustrating to deal with this as a reviewer.

Answer:

We apologise for the oversight

Action:

We have removed the repeated sections in the discussion.

  1. Lines 531-3: I don’t understand this section on supervision, so it may need rephrasing.

Answer:

Thank you for the comment

Action:

We have rephrased (lines 643-653).

As the comparative analysis has shown, there is a difference in approach between countries in relation to the presence of sanctions in the event of alcohol rules' transgression, the possibility for the worker to refuse to be tested, the obligation, or the simple possibility, of carrying out the alcohol tests in the workplace. The subjects responsible for monitoring compliance with the rules are different from country to country, also showing in this aspect a difference between the various members of the European Community. In fact, compliance with the rules in some cases is managed by the same companies (employer) while in other cases such as in Italy, for example, there are controls carried out by subjects with supervisory tasks on the laws relating to safety in the workplace who, in case of non-compliance, can directly raise criminal and administrative sanctions.

  1. Line 541-50 is a very long, multi-clause sentence which covers a number of themes. It is very hard to follow. The same is true for lines 553-60. The paper needs careful editing to iron out such overly-complicated sentences.

Answer

Thank you for pointing this out.

Action:

We decided to remove from line 541-550, the discussion was dealt with more clearly in the conclusions

For lines 553-660 we edited a new text (lines 661-668):

From our analysis, tests must be performed by a specialized health professional in Latvia; Italy seems to be the only country where the legislation, in addition to transferring the burden of controlling alcohol intake by employees, with an explicit mandate requires the occupational physician to also detect the absence of alcohol dependence in all workers who carry out activities in which they may be responsible for events that impact on the general population. This task appears to be very complex given the difficult detection of the problem of addiction through simple screening tests and in a therapeutic setting not suitable for the analysis of this kind of problem.

  1. Lines 567-8 strongly suggest that education and training are the most effective forms of prevention when it comes to alcohol. The evidence on prevention suggests otherwise: that education and training – unless done very well – can, in fact, be ineffective. I suggest this is either removed, or the authors familiarize themselves with more of the evidence on prevention.

Answer:

Thank you for pointing this out.

Action:

as suggested, we have preferred to remove it:

A particular need is then linked to the necessary training and information activities for workers, which is always the main way of prevention to be implemented in all contexts. The training must not only focus on issues on health damage resulting from reckless consumption of alcohol (in general and according to the specific work activity carried out), but also on the consequences in terms of work suitability and individual responsibility in the event of unsuccessful results. work test negatives.

  1. Lines 580-600 are hard to follow. As mentioned above, the discussion of dependence needs to be more precise (what does it mean to ‘shift into a state of real dependence’, for example?). Is the proposal to require MROs in all workplaces realistic? How much might it cost? What evidence is there to demonstrate MROs are effective in achieving these wider goals (beyond simply verifying the accuracy of test results)? Any evidence should be presented here, since the authors appear to be recommending this approach.

Answer:

Thank you for pointing this out.

Action:

we have preferred to remove it to make the discussion more focused on the primary topic.

The development of a specific European legislation and a detailed procedure about workplace policies against alcohol abuse in Europe could be a very important goal in the near future to stimulate also the countries still without a legal regulation about this important theme in both field of occupational health and public health. In this sense, it is considered important to foresee specific clinical and toxicological study paths (also in terms of type of tests and monitoring times) both in the case, obviously, of the finding of a state of alcohol dependence of the worker and for perhaps the most important case for a regular consumer (therefore more inclined to shift into a state of real dependence) than for the occasional consumer (very dangerous in terms of his own safety and that of third parties). What has just been pointed out appears to be an element of further caution towards third parties, in fact the monitoring of the problems through exclusively alcohol breath test appears - according to the writers - an insufficient measure. In order to better frame the problem and to better manage the alcohol risk, it would be advisable to hypothesize the introduction in Europe of the professional figure of the Medical Review Officer, a doctor appointed to interpret, verify and transmit the results of the tests carried out by the laboratory in the context of the company's alcohol and drug testing program [68–71]. In order to guarantee an effective service that complies with the highest scientific standards, the MRO must in fact have specific skills in clinical and analytical toxicology, clinical biochemistry, addiction medicine, forensic medicine and bioethics. In the USA, where this professional figure is now consolidated and has assumed a 'legal status', specific qualifying training courses are provided for MROs organized by scientific associations (American Association of Medical Review Officers, American College of Occupational and Environmental Medicine, American Society of Addiction Medicine, Western Institute for MRO Training) authorized to issue a certification enabling the practice of the profession recognized at national level [72–74] . The US legislation provides for a periodic 'retraining' of the MRO which is therefore required to follow a training program and to take a retraining exam every five years. specific qualifying trainings for MRO are provided for organized by scientific associations (American Association of Medical Review Officers, American College of Occupational and Environmental Medicine, American Society of Addiction Medicine, Western Institute for MRO Training) authorized to issue a certification enabling the exercise of profession recognized at national level [72–74].

  1. Lines 610-25 are also repeated material.

Answer: Thanking the reviewer we have preferred delete the part about the MRO to make the discussion more focused on the primary topic

Action:

Deleted part:

The US legislation provides for a periodic 'retraining' of the MRO which is therefore required to follow a training program and to take a retraining exam every five years. specific qualifying trainings for MRO are provided for organized by scientific associations (American Association of Medical Review Officers, American College of Occupational and Environmental Medicine, American Society of Addiction Medicine, Western Institute for MRO Training) authorized to issue a certification enabling the exercise of profession recognized at national level [72–74]. The US legislation provides for a periodic 'retraining' of the MRO which is therefore required to follow a training program and to take a retraining exam every five years. The Western Institute for MRO Training) authorized to issue a certification enabling the exercise of the profession recognized at national level. The US legislation provides for a periodic 'retraining' of the MRO which is therefore required to follow a training program and to take a retraining exam every five years. Western Institute for MRO Training) authorized to issue a certification enabling the exercise of the profession recognized at national level [72–74]. The US legislation provides for a periodic 'retraining' of the MRO which is therefore required to follow a training program and to take a retraining exam every five years.

  1. The conclusion suggests that a lack of homogeneity is a bad thing, but without explaining why. Is homogeneity necessary given the considerable variation in drinking profiles, prevalence and harms across the 27 countries? Importantly, are the authors proposing the standard approach should be strict (e.g. Latvia) or less intrusive. The arguments in this section need to be made much more carefully.

Answer:

Thank you for the suggestion

Action:

The conclusions were extended as follows (lines 692-714):

This legislative revision in the European context reveals a strong lack of homogeneity among the 27 countries of the Union on the issue of the management of alcohol-related problems in workers. Because the regulations of the individual states on health and safety in the workplace are drawn up on the basis of Community regulations, this lack of homogeneity appears incorrect. A common view of the problem is also necessary since the movement of workers between European countries would be easier in the case of similar rules. Otherwise, a worker could find oneself in such conditions that one could not cross a border because the blood alcohol level is different; this condition of non-homogeneity could lead that a worker may not practise in a neighbouring country until he has carried out specific clinical examinations.

The issue of controls relating to alcohol intake can also impact on the rights of the citizen and on the right to be able to refuse the test, so it seems strange that in Europe different legal conditions can be created between countries. A common vision of the problem could also lead the scientific community to study systems for the detection of conditions really at risk for workers and the community (establishing standard core exams).

Finally, the theme is of particular interest, since the problem of alcohol is so widespread in the population; the European Transport Safety Council claims on its website “We’re helping EU Member States share best practice on a wide range of road safety policies”; going beyond the topic of road safety and extending the concept to occupational safety our research group argue that a work of homogenization with the identification of minimum levels of caution to be shared at Community level would be appropriate for workers.

  1. Line 630: I suggest removing the word ‘endemic’ as it is poorly-defined and doesn’t reflect the heterogeneity of drinking cultures across, and within, countries

Answer:

Thanks for the clarification

Action:

We have removed the term “Endemic” and rephrased the whole text (see point 19 reviewer n. 2).

  1. I think the conclusion needs to return to key issues around e.g. implementation (what might the practical implications of adopting workplace testing be), human rights (what are the implications of random testing or summary dismissal for the rights of workers), levels of severity across different regimes, and the difference between banning alcohol from the workplace and testing for drunkenness. At the moment it only briefly sets out the case for a more standardized approach.

Answer:

Thanks for the comment.

Action:

The conclusions were extended (see point 19 reviewer n. 2).
